# THAP9 Transposase Cleaves DNA via Conserved Acidic Residues in an RNaseH-Like Domain

**DOI:** 10.3390/cells10061351

**Published:** 2021-05-29

**Authors:** Vasudha Sharma, Prachi Thakore, Sharmistha Majumdar

**Affiliations:** Discipline of Biological Engineering, Indian Institute of Technology Gandhinagar, Gujarat 382355, India; vasudha.sharma@iitgn.ac.in (V.S.); prachi.t@iitgn.ac.in (P.T.)

**Keywords:** catalytic triad, DNA transposition, DDE, RNaseH domain

## Abstract

The catalytic domain of most ‘cut and paste’ DNA transposases have the canonical RNase-H fold, which is also shared by other polynucleotidyl transferases such as the retroviral integrases and the RAG1 subunit of V(D)J recombinase. The RNase-H fold is a mixture of beta sheets and alpha helices with three acidic residues (Asp, Asp, Glu/Asp—DDE/D) that are involved in the metal-mediated cleavage and subsequent integration of DNA. Human THAP9 (hTHAP9), homologous to the well-studied Drosophila P-element transposase (DmTNP), is an active DNA transposase that, although domesticated, still retains the catalytic activity to mobilize transposons. In this study we have modeled the structure of hTHAP9 using the recently available cryo-EM structure of DmTNP as a template to identify an RNase-H like fold along with important acidic residues in its catalytic domain. Site-directed mutagenesis of the predicted catalytic residues followed by screening for DNA excision and integration activity has led to the identification of candidate Ds and Es in the RNaseH fold that may be a part of the catalytic triad in hTHAP9. This study has helped widen our knowledge about the catalytic activity of a functionally uncharacterized transposon-derived gene in the human genome.

## 1. Introduction

Transposons are mobile DNA sequences that can move from one genomic location to another using autonomous or non-autonomous enzyme machinery [1]. In the course of their propagation in the host genome, these selfish genetic elements are known to disrupt host cellular functions by ectopic recombination events, insertion mutations and interference with gene regulation [2,3,4]. In some cases, entire transposons or parts of the same are co-opted or ‘domesticated’ by the host to serve beneficial functions. Numerous genes have recruited promoters, enhancers, DNA binding domains, alternative splice sites, polyadenylation sites and cis-regulatory sequences from an ancestral DNA transposon [5] to make evolutionarily beneficial, functional host genes [2,4].

A DNA transposon, also known as a ‘cut and paste’ transposon, uses a transposase enzyme, often encoded by itself, to cut or excise itself from one location in the genome and paste or integrate it in a different genomic location, without the need of homology between donor and target DNA sequences [6]. DNA transposons contribute to 3% of the human genome and around 50 human genes are derived from these elements [7]. TE-derived genes may become immobile and exist as single copies by losing the hallmarks of transposons, i.e., terminal inverted repeats (TIRs) and target site duplications (TSDs) [5]. RAG1/2, CENPB, SETMAR and hTHAP9, which are examples of human genes that have domains recruited from DNA transposons [2], have evolved to have diverse functions. For example, RAG1/2 mediates the DNA rearrangement reactions in V(D)J recombination; CENPB (Centromeric protein B) helps in chromatin assembly at centromeres during cell division; SETMAR is a DNA repair protein while the function of hTHAP9 is still unknown.

All 17 eukaryotic cut-and-paste transposase superfamilies have a D, D, E/D (Asp, Asp, Glu/Asp) catalytic triad in the active sites of their respective transposases [8]. The third residue in the triad can be either a D or E; hence, it is represented as a DDE/D motif. Topologically, the DDE/D triad is always present in an ‘RNase-H like fold,’ which is a mixture of alpha helices and beta sheets [9]. The conserved RNase-H fold has a characteristic secondary structure—β1-β2-β3-α1-β4-α2/3-β5-α4-α5—wherein the first D of the triad is always present on β1, the second D on or just after β4 and the E/D either on or just before α4 [6]. The RNase-H like fold is present in many transposases (Tn3, Tn5, Mu, hAT, Tc1/Mariner) and retroviral integrases (HIV-1 integrase, RSV integrase), in which the DDE/D motif catalyzes magnesium ion-dependent nucleophilic attack on DNA [9,10,11].

Wessler et al., in 2011, predicted the DDE motif in P superfamily; this has been confirmed in the recently solved cryo-EM structure of *Drosophila* P-element transposase (DmTNP) [8,12]. The P-element is a widely studied transposon in *Drosophila melanogaster* that invaded this species recently and is continuing to spread through the natural fly populations by horizontal gene transfer [13]. It was discovered that P-element transposition is responsible for hybrid dysgenesis leading to male sterility in *Drosophila* [14]. Ever since their discovery, P-elements have been used as a model to study transposition and as a genetic and genome engineering tool [15,16]. The DmTNP cryo-EM structure has revealed that the RNase H domain is located near the donor-target DNA junctions, with the catalytic residues coordinating a Mg^2+^ ion. GTP is not required for providing energy for the reaction; rather, it possibly positions transposon DNA for catalysis by directly interacting with the terminal base of the cut transposon. Other unique features highlighted in the DmTNP transpososome structure are the looping out of the cut strand to maintain DNA integration and the bending of target DNA [12,17].

The hTHAP9 protein is homologous (25% identical and 40% similar) to the DmTNP. It is present as a single copy in the human genome and lacks transposon hallmarks, i.e., TIRs and TSDs; this suggests that human THAP9 may have undergone molecular domestication and is immobile, whereas its homolog in zebrafish (Pdre2) appears to be functional since it has intact TIRs and STIRs and multiple copies in the zebrafish genome [16,18,19]. Although the function of hTHAP9 remains undiscovered, it is catalytically active and can cut DNA flanked by P element’s TIRs in humans as well as *Drosophila* cells [20]. Since hTHAP9 has retained its ability to cut DNA, it is important to investigate the nature of its catalytic domain. Using secondary structure predictions, homology modeling and multiple sequence alignment, we identify an RNase-H like fold in hTHAP9. Furthermore, we experimentally demonstrate that D304, D374 and E613 are the catalytic residues of hTHAP9 and are responsible for DNA excision.

## 2. Materials and Methods

### 2.1. Materials

HEK293 cell lines obtained from Dr Virupakshi Soppina’s lab at IIT Gandhinagar, DMEM (HyClone, Logan, UT, USA) supplemented with 10% Fetal Bovine Serum, TurboFect Transfection Reagent (Thermo Fisher Scientific, Waltham, MA, USA; R0531). pISP2/Km, Cg4-Neo, pBluescript vectors were generous gifts from Prof. Donald Rio, UC Berkeley.

G418 (Merck, Darmstadt, Germany, 345810), GeneJET plasmid miniprep kit (Thermo Fisher Scientific, Waltham, MA, USA; K0503), Ampicillin (Sigma, St. Louis, MO, USA; 10835242001), Kanamycin (Sigma, St. Louis, MO, USA; K1377), Restriction Enzyme PvuII-HF (NEB, Ipswich, MA, USA; R0151S), CutSmart Buffer (NEB, Ipswich, MA, USA; B7204S), rabbit anti-HA monoclonal antibody (Sigma, St. Louis, MO, USA; SAB1306169), mouse anti-GAPDH monoclonal antibody (Novus Biologicals, Littleton, CO, USA; NB300-328SS), anti-rabbit (GENA934) and anti-mouse (GENA931) IgG HRP secondary antibodies.

### 2.2. Methods

#### 2.2.1. Multiple Sequence Alignment and Secondary Structure Prediction

hTHAP9 (Uniprot Identifier: Q9H5L6-1), DmTNP (Uniprot Identifier: Q7M3K2-1) and Pdre2 [18] amino acid sequences were aligned using Clustal omega [21] with default parameters and the results were viewed in MView [22]. PSIPRED [23] was used with default parameters to predict the secondary structure of the three THAP9 homologs.

#### 2.2.2. Homology Modeling, Docking and MD Simulations

Homology modeling was performed using default parameters and target sequence-template option in the freely available SWISS MODEL [24] with a DmTNP (6p5a) structure as template. The output from the SWISS MODEL was used to perform magnesium ion docking using AUTODOCK Vina [25]. The hTHAP9 homology model with magnesium ion docked in it was further used for MD simulations with GROMACS [26]. The model was processed using CHARMM 36 force field [27] and further simulated for 100 ns. The final structure was viewed and analyzed using PyMol [28]. RMSD was calculated using GROMACS and the Ramachandran plot was analyzed using SAVES software [29].

#### 2.2.3. Site-Directed Mutagenesis

Primers for generating mutations in hTHAP9 and DmTNP coding sequences were designed using NEB Base changer tool^TM^ and site-directed mutagenesis was performed using the Q5 Site-directed Mutagenesis kit (NEB, Ipswich, MA, USA E0554S) as per the manufacturer’s instructions. All mutant constructs were verified by Sanger sequencing at Scigenom, India.

#### 2.2.4. In Vivo P Element Excision Assay

HEK293 cells (0.5 × 10^6^, grown in DMEM supplemented with 10% fetal bovine serum under standard tissue culture conditions) were plated per well of 6-well plates and transfected at 70–90% confluency using Turbofect transfection reagent. The standard transfection protocol of the manufacturer was followed to transfect 1 ug pISP2/Km reporter [20] plasmid (which has 0.6 kb P element insertion, which disrupts the kanamycin resistance open reading frame) along with either 1 ug DmTNP or hTHAP9 (wild type or mutants separately) or pBluescript empty vector (negative control), in triplicates. After 48 h, the cells were harvested in PBS to recover and analyze plasmid DNA and protein expression. The plasmid DNA from the transfected cells was isolated using GeneJET plasmid miniprep kit following the standard protocol. The DNA was quantified, and 1 µg of total DNA was transformed into *Escherichia coli* DH5α^™^ ultracompetent cells (RecA-) and was plated on ampicillin and kanamycin + ampicillin plates. In total, 1/10 of the cells were plated on ampicillin plates while 1/2 of the cells were plated on kanamycin plus ampicillin plates. After 24 h, the colonies were counted.

The ratio of the number of kanamycin-ampicillin colonies to that of the ampicillin colonies gave the P element excision activity [20]. The relative excision activity was calculated using the following formula:Excision activity = (Number of KanR + AmpR colonies)/Total number of AmpR coloniesRelative Excision activity= (Excision activity of mutant × 100)/Excision activity of wild type transposase

Statistical significance of relative excision activity values was calculated using a two-tailed non-parametric t-test followed by Mann–Whitney’s test.

Plasmid DNA (pISP-2/Km reporter [20]) was isolated from the colonies obtained on the kanamycin plus ampicillin plates. pISP2/Km, which is originally kanamycin-sensitive, gives rise to kanamycin-resistant colonies due to P element excision and subsequent repair. pISP2/Km is a 4483 bp plasmid with a 600 bp P element between 2240 and 2840 positions. PvuII restriction endonuclease cuts pISP2/Km at positions 2054, 3173, 3935 and 4173 [20].

In total, 1 µg of isolated repaired reporter DNA was digested with PvuII as per the standard restriction digestion protocol. If P element excision has taken place, PvuII digestion of the repaired pISP2/Km reporter plasmid will give rise to a 0.5 kb fragment. On the other hand, if excision has not occurred, a 1.1 kb fragment (made up of 0.6 kb P element and 0.5 kb of kanamycin resistance gene) is observed with three other bands corresponding to 2 kb, 0.75 kb and 0.25 kb (Appendix A).

To investigate the nature of donor site repair after P element cleavage, the isolated reporter plasmids were sequenced across the P element excision site using a primer (5′-GTTGTGTGGAATTGTGAGCGG-3′) flanking the P element insertion site.

#### 2.2.5. In Vivo Integration Assay

Cg4-Neo reporter plasmid [20] (which carries an SV40 promoter-G-418R gene inside the Cg4 P element vector) was co-transfected (Turbofect), with the transposase expression plasmid (DmTNP or hTHAP9 wild type or mutants separately) or negative control (pBluescript or pCDNA3.1 (+) empty vector) in HEK293 cells (0.5 × 10^6^ cells per well of a 6-well plate) at 70% confluency. A total of 2 ug of plasmid DNA was transfected per well: 50 ng of CgNeo4, 1 ug of the transposase plasmid/negative control and 950 ng pBluescript empty vector. After 48 h, the cells from each well were transferred to separate 10 cm dishes and were allowed to adhere for 24 h at 37 °C in a CO_2_ incubator. After 24 h, the cells were grown in media supplemented with G418 (0.5 mg/mL) and selected for 2 weeks, after which the G-418-resistant colonies were fixed with methanol, stained with crystal violet and counted. The relative integration activity was computed as a percentage of activity of each mutant to the wild type DmTNP [20]. Significant values were determined using a one-way ANOVA method followed by Dunn’s test.

#### 2.2.6. Protein Immunoblotting

The cell extracts from transfected cells were lysed and run on 10% denaturing SDS PAGE in running buffer (1X Tris- Glycine SDS). PageRuler™ Prestained Protein Ladder (10 to 180 kDa, ThermoFisher Scientific, Waltham, MA, USA; 26616) was used as a molecular size marker and the samples were transferred to PVDF membrane by electroblotting using a standard gel transfer system. The membrane was blocked to remove nonspecific binding using 3% (*w*/*v*) skimmed milk in Tris-Buffered Saline with Tween-20 (TBST). Membranes were washed with 1X TBST and incubated overnight at 4 °C with 1:2000 dilution of primary anti-HA antibody followed by incubation with 1:5000 dilution of HRP- coupled secondary antibodies for 2 h at room temperature and detected using enhanced chemiluminescence (Pierce, PI32106) on a Bio-Rad Gel Documentation system.

## 3. Results

The catalytic domains of all known eukaryotic cut-and-paste transposases have a characteristic RNase-H-like fold with a conserved DDE/D motif. The well-studied Drosophila P element transposase (DmTNP) is a member of the P superfamily, which is one of the 17 known cut-and-paste TE superfamilies [8,12]. hTHAP9, a homolog of DmTNP, was reported to be an active transposase that can cut P-element DNA flanked with intact inverted repeats [20]. Thus, we decided to investigate if hTHAP9 had a RNaseH fold-like catalytic domain [8].

### 3.1. Prediction of Secondary Structure Elements and RNase-H domain in hTHAP9 and Its Homologs

Many DDE/D transposases, including DmTNP and retroviral integrases, have similar catalytic domains [30]. The DDE transposases always have an RNase-H fold in their catalytic domain, within which the first D lies in β1, the second D in or just after β4 and the third E/D in α4 [8,30].

Using PSIPRED [23], we predicted the secondary structure elements present in full length DmTNP, hTHAP9 and Pdre2. The presence of RNase-H fold was inspected visually by locating the pattern of β sheets and α helices in the PSIPRED output. The Ds and E that were found in beta sheets (β1 and β4) and alpha helices (α4) are in agreement with the characteristic structure of RNase-H fold in integrases and other DDE/D transposases.

In DmTNP, the reported residues of the catalytic triad, namely D230, is present in β1, D303 in β4 and E531 in α4 (Figure 1). For hTHAP9 and Pdre2, we have predicted the catalytic triad by comparing the predicted secondary structure to the well-studied RNase-H fold structure, common in transposases (Rag1, Hermes, Mos1, DmTNP) and integrases (HIV1). We have identified D304 in β1, D374 in β4 and E613 in α4 in hTHAP9. Similarly, we found D367 in β1, D374 just after β4 and E679 in Pdre2 (Figure 1). The RNase-H folds in DmTNP, hTHAP9 and Pdre2 appear to be disrupted by an insertion domain between β5 and α4. In DmTNP, the insertion domain has been reported to carry a unique GTP binding domain [12], but the role of insertion domains in hTHAP9 and Pdre2 are unknown.

### 3.2. Homology Modeling of hTHAP9 Identified a Putative RNaseH-Like Domain

To further investigate if the secondary structure elements predicted in hTHAP9 were indeed able to fold into an RNaseH-like domain, we made a 3D model of hTHAP9 using the recently solved DmTNP Cryo-EM structure (6p5a) as a template for homology modeling. SWISS-MODEL [24] interactive GUI was used for generating the hTHAP9 model with default parameters and the output structure was docked with magnesium ion using AUTODOCK [25]. We then performed energy minimization for the docked structure using MD simulations for 100 ns with GROMACS [26] and CHARMM 36 force field [27]. The resultant energy minimized model was viewed in PyMol and analyzed using SAVES. In total, 98.2% of the total residues were in allowed regions and only 0.5% in disallowed regions, as revealed by the Ramachandran plot analysis (Appendix A).

As predicted in Figure 1, D374 and E613 are, respectively, present in a beta sheet (β4) and α-helix (α4) within the RNaseH fold (Figure 2). The two residues are also proximal to the magnesium ion, forming an active site. Although D304 appeared just after a beta sheet, it was found to be away from the other catalytic residues. As predicted by the SWISS-MODEL secondary structure, D304 lies in the loop region immediately after a β sheet of the final 3D model (Appendix A). It is possible that due to low sequence similarity of hTHAP9 with the template, many predicted residues did not form a proper conformation and ended up in loops. Interestingly, this distal position of hTHAP9’s third acidic residue (D304) away from the catalytic pocket resembles the position of E962 in the catalytic triad of Rag1 recombinase [31]. A conformational change in Rag1 during catalysis brings E962 (third acidic residue) proximal to the catalytic site [31].

### 3.3. Identification of Other Conserved Acidic Residues in the Catalytic Domain of hTHAP9 and Its Homologs 

We used multiple sequence alignment to examine the conserved acidic residues in *Drosophila* P element transposase (DmTNP), Human THAP9 (hTHAP9) and Zebrafish THAP9 (Pdre2). Many residues throughout the sequences, including several Ds and Es, were conserved among the three homologs (Figure 3, marked with black boxes).

### 3.4. Signature Stringin the RNaseH-like catalytic domain

Besides the canonical DDE motif, additional conserved residues in the RNaseH-like catalytic domain have been reported to form a signature string specific to each ‘cut and paste’ transposase superfamily [8]. The residues in a signature string have similar approximate spacing in all the members of the superfamily. For the P superfamily, the signature string (Figure 4), first described by Wessler et al. in 2011, carries the catalytic triad (D230, D303 and E531) as well as a [C/D](2)H motif (D339) downstream of the second D of the DDE triad. This motif is always right after β5 in the predicted secondary structure. We have identified a similar pattern of conserved amino acids in hTHAP9 and Pdre2 by visually inspecting the sequences in the multiple sequence alignment results (Figure 4).

The conserved Ds and Es in the RNase-H fold of the three homologs are highlighted in Figure 3 and listed in Table 1.

### 3.5. Site-Directed Mutagenesis of Conserved Acidic Residues in the Catalytic Domain of hTHAP9 and Its Homologs

To investigate the role of the conserved acidic residues (Table 1 and Figure 3), we mutated all the conserved Aspartate (D) residues to Cysteine (C) and the Glutamate (E) residue to Glutamine (Q). We avoided the conventional alanine (A) substitutions because the size of A is very small in comparison to D and E; replacement with a smaller amino acid may affect the protein structure, and thus, it will be difficult to infer whether potential loss of activity is due to the absence of catalytic activity or structure disruption [32]. Moreover, since Metnase/SETMAR is known to function using a DDN catalytic triad, we decided not to substitute D → N, since it may not affect the catalytic activity of the transposase [33]. Thus, in order to replace D or E with a similar-sized amino acid, we substituted them with a C or Q, respectively [34,35]. The mutants cloned in pCDNA 3.1 (+) include hTHAP9 (D304C, D374C, E776Q, E613Q, D414C, D519C, D695C) and DmTNP (D303C, D230C, D444C, D339C, D252C, E531Q, D604C, D676C).

### 3.6. Many Conserved Acidic Residues in the Catalytic Domain of hTHAP9 and Its Homologs Are Important for DNA Excision

To investigate the catalytic activities of the hTHAP9 and DmTNP mutants, we used a plasmid-based DNA excision assay [20]. In this assay, a reporter plasmid (pISP-2/Km, having 0.6 Kb insertion of P element DNA flanked by intact terminal inverted repeats disrupting the Kanamycin resistance gene) is co-transfected with a plasmid encoding for the transposase (Figure 5A). We ensured that all the mutant THAP9 proteins were expressed in human cells in approximately equal amounts (Appendix A).

The excision activity of each mutant (Appendix A) was measured by scoring transposase-induced transposon excision events in bacteria (described in the methods section, [36]). We further confirmed P element excision, by sequencing (Appendix A) or performing diagnostic PvuII restriction digestion (Appendix A) of plasmid DNA excision products recovered from bacteria. When the P element DNA (600 bp) is excised by the functional transposase followed by repair of the reporter plasmid, the PvuII restriction pattern of the recovered reporter plasmid shifted by 600 bp (Appendix A).

We observed that in DmTNP, D230C, D303C and E531Q exhibit the lowest relative excision activity (~10% of the wildtype DmTNP activity) (Figure 5). These residues have been previous reported to constitute the DDE motif in DmTNP [8,12]. Besides the DDE triad, we also found that D444 is extremely important for DmTNP transposition because the number of kanamycin resistant colonies observed in all replicates was zero (Figure 5, Appendix A). Moreover, D339 from the conserved D(2)H motif, which is a part of the signature string [12] (Figure 4) of the P superfamily, also affects the excision activity when mutated (Figure 5).

In hTHAP9 (excision activity of wild type hTHAP9 was observed to be 70% of that of wild type DmTNP), the relative excision activities were observed to be low for the D304C, D374C and E613Q mutants. This result is in agreement with our computational prediction of the catalytic triad. Surprisingly, the excision activity of D695C was enhanced in comparison to the wild type protein (Appendix A).

### 3.7. Many Conserved Acidic Residues in the Catalytic Domain of hTHAP9 and Its Homologs Are Important for DNA Integration

We next investigated whether the mutant transposases could carry out transposition of a marker gene flanked by P element terminal inverted repeats, from a plasmid into the human genome. The relative integration activity (Appendix A) was calculated by screening and staining G418-resistant colonies (described in the methods section, [20]).

In DmTNP, mutation of catalytic triad DDE residues, D230, D303 or E531, led to low DNA integration activity (Figure 6). The D444C mutant underwent complete loss of excision as well as integration function, suggesting its role in DNA–protein interactions during transposition (Figure 6). Consistent with the DNA excision results (Figure 5), mutations of D304, D374 and E613 in hTHAP9 resulted in low DNA integration activity. Surprisingly, the D695C mutant, which showed heightened excision assay (Figure 6), demonstrated low integration activity (Figure 6, Appendix A).

Thus, we identified the catalytic domain of hTHAP9 and demonstrated that it has a putative RNase-H like fold (like in DmTNP) with several conserved acidic residues, including D304, D374 and E613, that may constitute the catalytic triad (D230, D303 and E531 in DmTNP) and are important for DNA excision and integration.

## 4. Discussion

The architecture of most primitive proteins, as deduced by molecular fossils found in modern proteins, share a handful of folds, namely P-loop, TIM β/α-barrel, Rossmann-fold domains, Ferredoxin-like, Flavodoxin-like and Ribonuclease H-like folds [37]. The RNase-H fold, which is one of the most ancient and abundant protein folds derived from viruses, has been co-opted by diverse proteins in all kingdoms of life (transposases, integrases, resolvases, Piwi nucleases, CRISPR-associated enzymes such as Cas9, RAG1 etc.) [37,38] that are involved in a variety of cellular processes, such as DNA replication, recombination and repair, transposition, immune defense, splicing, RNA interference and CRISPR-Cas immunity [38].

The members of all 17 eukaryotic cut-and-paste transposase superfamilies (including pogo, Tc1/Mariner, Mutator, Harbinger, Merlin, PIF, Zator, Sola, Ginger, Transib, piggyBac, hAT [35,39,40,41,42,43,44]) carry the characteristic RNase-H fold [8], suggesting a common evolutionary origin of the “cut-and-paste” transposition mechanism [11,45,46]. The RNase-H fold is also found in viral integrases (HIV-1, RSV [11]) that share strikingly similar structural properties to several transposases, LTR transposons (Copia, Gypsy) and bacterial IS sequences (e.g., IS1, IS3, IS6, IS30) [6,47,48], suggesting a common ancestor [11,49,50].

Although the function as well as overall architecture of RNase-H fold-containing proteins is very diverse, the architecture of their RNaseH-like catalytic core and the corresponding catalytic mechanism is highly conserved [49,51]. The RNase-H fold brings three catalytic residues (DDD/E) in close proximity, which coordinate with two divalent metal ion cofactors (mostly Mg^2+^) to form the active site for both DNA cleavage and strand transfer during transposition [6,46]. The conserved steps during catalysis include initial DNA cleavage by hydrolysis, formation of an oligomeric synaptic complex that carries out catalysis in trans (i.e., the enzyme subunit will bind one transposon end and cut the opposite end of the transposon DNA), the subsequent strand transfer that involves the nucleophilic attack on target DNA by the free 3’-OH at the transposon end and strong bending of target DNA [6].

RNase-H fold-containing proteins are multi-domain proteins which often carry other domains involved in site-specific DNA binding and multimerization in addition to the RNaseH-like catalytic domain. The DDE transposase family members studied to date exhibit striking differences in their transposition mechanisms. For example, the oligomeric state of the synaptic complexes that are formed by various transposases are quite different (DmTNP, Tn5 and Mos1 are dimeric, Mu and PFV are tetrameric, while Hermes transposase is octameric). Additionally, different transposases have unique mechanisms of second strand cleavage [12,30,36,50,52].

Moreover, both the size of the overall domain as well as individual secondary structure elements of the RNaseH-like catalytic domain vary significantly. This is because the RNase-H fold of many DDE/D transposases contain an insertion domain [6] often inserted between the fifth β-strand and fourth α-helix. The length and function of the insertion domain varies in different DDE transposases, as observed in the limited number of corresponding solved structures (Tn5, Hermes, Rag1, DmTNP) [11,17,30]. Although the exact functions of these insertion domains have not been determined, it is observed that they often contain amino acids essential for transposition function. For example, in Tn5 transposase, the insertion domain is important for formation of a DNA hairpin at the transposon ends, while the insertion domain of Hermes transposase carries a conserved tryptophan residue (Trp319) responsible for target joining. The insertion domain of DmTNP harbors an alpha-helical GTP binding domain essential for binding GTP as a prerequisite for successful DNA transposition [12,17]. It is also interesting to note that despite the dissimilarity in primary sequence and functions, the insertion domains of Hermes, Rag1, Tn5 and DmTNP are topologically similar.

Our study predicts the presence of an RNase-H fold in hTHAP9 (and Pdre2), which appears to be disrupted by an insertion domain between β5 and α4, similar to the insertion domain of DmTNP which binds GTP via GTP-binding motifs [12,17]. Despite the disruption of the RNase-H fold, both DmTNP and hTHAP9 are able to cut the substrate DNA [20].

The DDE/D motif has been shown to be functionally relevant in several transposases by investigating catalytic activity after performing site-directed mutagenesis to substitute the essential DDE triad with other amino acids [6]. Previous studies had defined the DDE motif in DmTNP (namely D230, D303 and E531) by substituting each of the acidic residues in the triad with alanine and demonstrating that these mutants had significantly lower DNA excision activity [12]. However, alanine is much smaller than D/E amino acids and may have disrupted the RNaseH fold. To validate that excision is indeed decreasing because of loss of essential catalytic amino acids and not fold disruption, we replaced the motif with amino acids of comparable sizes. We observed that D230, D303 and E531 fail to excise and integrate the target DNA when mutated to Cys (Figure 5 and Figure 6). In hTHAP9, DNA excision and integration were severely impaired when residues D304, D374 and E613 were mutated. This suggests that these three residues, which are conserved in its homologs (DmTNP, Pdre2) and are located on appropriate secondary structure elements in the predicted RNaseH fold, may form the DDE motif in hTHAP9. Moreover, it was observed that besides the predicted catalytic triad, there were other conserved acidic residues in the signature string of hTHAP9, which were important for DNA transposition (Appendix A). These include D414 (D339 in DmTNP), which is a part of the [C/D](2)H motif, and D519 (D444 in DmTNP). D339 marks the boundary of the insertion domain (residues 339–528) of DmTNP [12]. Additionally, D444 is absolutely essential for the transposition function (Figure 5 and Figure 6, Appendix A); it is interesting to note that in the recently solved DmTNP structure, D444 mediates binding to GTP, which is coordinated to a magnesium ion [12].

It is interesting to speculate about the exact roles of other conserved residues and motifs in the catalytic domain of hTHAP9 and its homologs. The recent cryo-EM structure of DmTNP [12] illustrates that the catalytic domain makes extensive contact with both DNA and GTP during transposition. Thus, future biochemical and structural investigations of hTHAP9 may identify additional conserved residues involved in various steps of the transposition process as well as confirm the identity of the hTHAP9 catalytic triad.

The hTHAP9 D695C mutant displayed elevated transposase activity that was higher (1.4 times) than the wild type hTHAP9 activity (Figure 5, Appendix A). However, its DNA integration activity was not enhanced to a similar extent (Figure 6). Hyperactive mutants of *Sleeping beauty* (SB100X) and *Piggy-Bac* (hyPBase) transposons are used as gene delivery vectors for stable expression of transgenes [53,54]. The hTHAP9 D695C mutant may also serve as an attractive genome engineering tool upon further investigations.

## 5. Conclusions

This study highlights the critical importance of conserved acidic residues in the RNaseH-fold containing catalytic domain of human THAP9. The RNaseH fold of hTHAP9 was predicted using homology modeling and secondary structure predictions. Mutagenesis of conserved acidic residues in the catalytic triad (DDE/D) as well as the canonical signature string of the predicted RNaseH fold significantly impacted catalytic function i.e., DNA excision and subsequent integration during transposition. Moreover, we identified an Asp residue which when mutated can achieve heightened excision activity; such hyperactive transposases can be exploited as possible genome engineering tools. Thus, this study provides new insights towards elucidating the catalytic mechanism and unknown physiological roles of a relatively unexplored human transposase.

## Figures and Tables

**Figure 1 cells-10-01351-f001:**
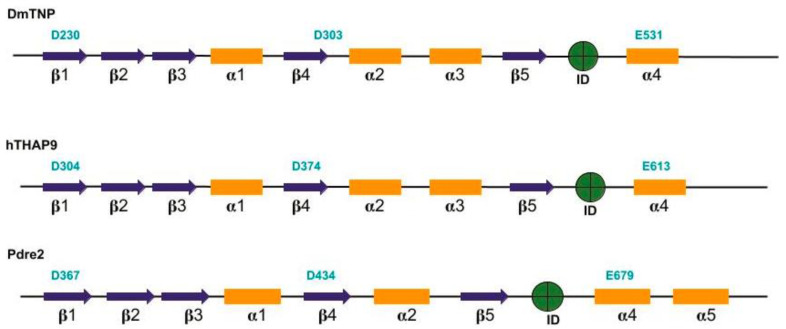
Predicted RNase-H fold in Drosophila, human and zebrafish THAP9. DDE motifs for each homolog are represented in bold (cyan) above its putative location. Blue arrows indicate beta sheets and yellow bars represent alpha helices. Insertion domains for each homolog (DmTNP (residues 339–528) [12], hTHAP9 (residues 415–604), Pdre2 (residues 475–670)) are shown in green circles.

**Figure 2 cells-10-01351-f002:**
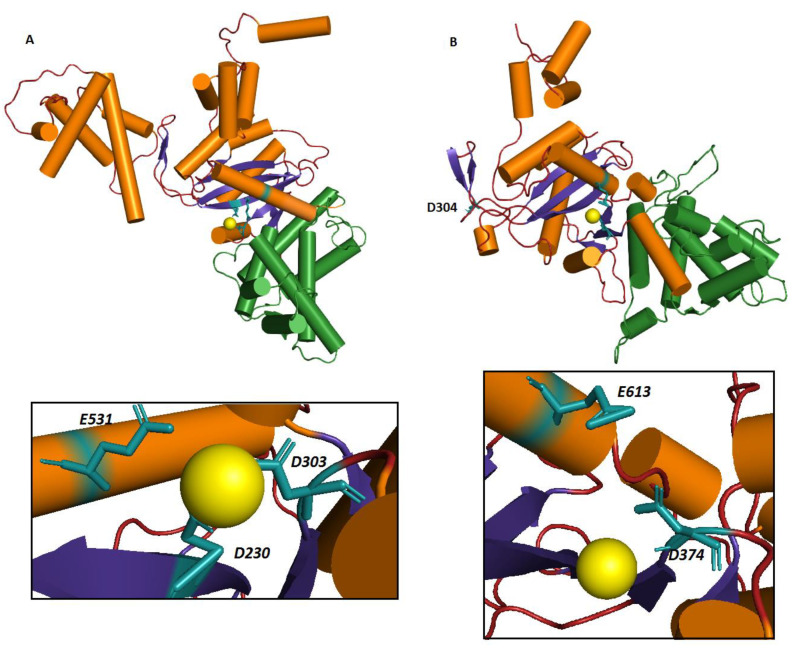
Predicted catalytic triad shown in the 3D structure of hTHAP9, built using homology modeling. (**A**) DmTNP structure with beta sheets (blue arrows), helices (orange rods) and insertion domain (green) (PDB ID-6P5A(12)). The active site is shown in the inset with DDE motif (D230, D303, E531) in close proximity to the magnesium ion (light yellow). (**B**) Homology model of hTHAP9 with the magnesium ion. Active site shown in the inset has D374 and E613 close to the magnesium ion (yellow). D304 does not adopt any secondary structure but follows immediately after a beta sheet.

**Figure 3 cells-10-01351-f003:**
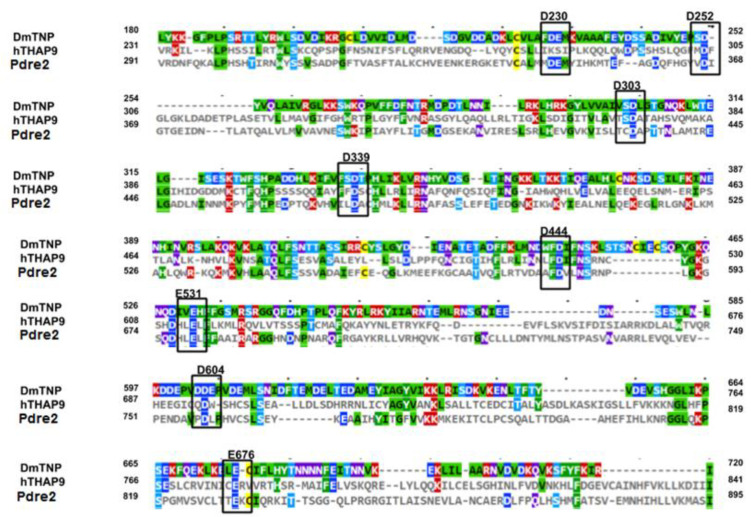
Sequence conservation in the putative RNaseH-domains of DmTNP, hTHAP9 and Pdre2. Amino acid sequences of DmTNP, hTHAP9 and Pdre2 were aligned using Clustal Omega and were visualized using MView. The Ds and Es that were conserved in the putative RNaseH-domains of all three homologs are highlighted in black boxes and labeled (in bold on top of the sequence) with the corresponding DmTNP residue number. The coloring scheme of the alignment is based on conserved physicochemical properties of amino acids (Colored by identity-Palette P1 of MView, refer to Appendix A for the key).

**Figure 4 cells-10-01351-f004:**
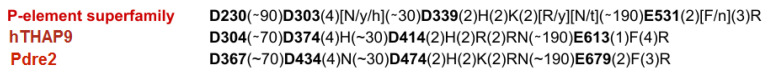
Signature string of P-element superfamily in hTHAP9 and Pdre2. Signature string of the P element superfamily [8] along with similar signature strings identified in human and zebrafish THAP9. Conserved Ds and Es are represented in bold and the numbers represent the spacing between conserved amino acid residues.

**Figure 5 cells-10-01351-f005:**
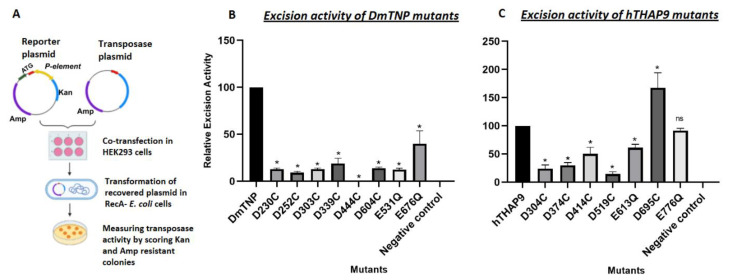
Conserved acidic residues within the catalytic triad and signature string of DmTNP and hTHAP9 are important for DNA excision. (**A**) Schematic of transposon excision assay. Relative excision activities of (**B**) DmTNP mutants (compared to wild type DmTNP) and (**C**) hTHAP9 mutants (compared to wild type hTHAP9). Negative control (pBluescript empty vector). All the values are represented as a mean of four independent experiments (±SEM, *n* = 4) *p* value < 0.05, * (*p* < 0.033), ns (*p* < 0.12).

**Figure 6 cells-10-01351-f006:**
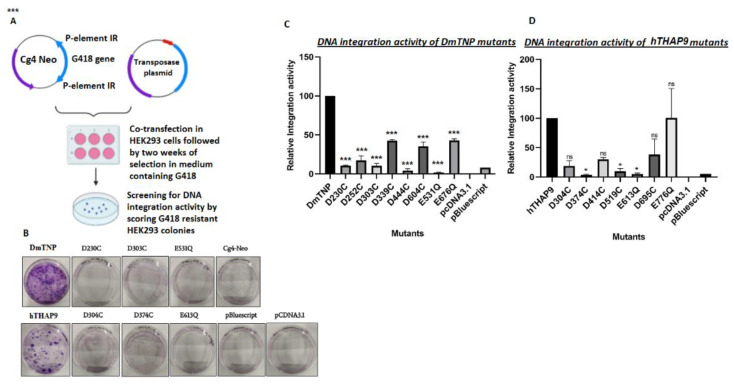
Conserved acidic residues within the catalytic triad and signature string of DmTNP and hTHAP9 are important for DNA integration. (**A**) Schematic of the transposon integration assay. (**B**) Crystal violet-stained G418-resistant colonies of wildtype Drosophila and hTHAP9 along with DDE motif mutants and negative controls. Relative integration activities of (**C**) DmTNP mutants (compared to wild type DmTNP) and (**D**) hTHAP9 mutants (compared to wild type hTHAP9). Empty vectors pCDNA3.1 (+) and pBluescript were used as negative controls. All the values are represented as the mean of two independent experiments performed in duplicates (±SEM). *p*-value < 0.05, *** (*p*<0.001), * (*p*<0.033), ns (*p* < 0.12).

**Table 1 cells-10-01351-t001:** List of conserved Asp and Glu residues (which were mutated) in the RNase-H fold of DmTNP and hTHAP9. Residues of each homolog, which are aligned with each other in Figure 3, are adjacent to each other in the same row.

DmTNP	hTHAP9
D230	
D252	D304
D303	D374
D339	D414
D444	D519
E531	E613
D604	D695
E676	E776

## Data Availability

Not applicable.

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
