# Peer review of "THAP9 Transposase Cleaves DNA via Conserved Acidic Residues in an RNaseH-Like Domain"

_cells, 2021, doi:10.3390/cells10061351_

Round 1
Reviewer 1 Report
In this manuscript, Sharma et al characterize a human transposase hTHAP9, which is homologous to the well-studied Drosophila P-element transposase. Retro transposase typically possesses RNase-H fold in their structure, which is a mixture of beta sheets and alpha helices with three acidic residues (Asp, Asp, Glu/Asp - DDE/D motif) that are involved in the metal-mediated cleavage and subsequent integration of DNA. First, they modeled hTHAP9 using cryo EM structures of DmTNP to identify RNase-H like fold along with important acidic residues in its catalytic domain. The authors then performed site-directed mutagenesis of the predicted catalytic residues and screened for DNA excision and integration activity. They used an established reporter assay system for this purpose. They found that candidate Ds and Es in the RNaseH fold forms the part of the catalytic triad in hTHAP9. Additionally, the authors found a residue (D695C), which when mutated, led to an increase in hTHAP9’s transposition activity.
This is a well-executed study and a well-written manuscript. Some points to improve the manuscript are given below.
- The abstract is written with subheadings (background, results, conclusions). This is unnecessarily descriptive. Authors should present an abstract in a concise form without these obvious subheadings.
- 5 and Fig 6C: Does the statistical significance apply to all the mutants in the panel equally? It is more intuitive to show the test results individually, like in Fig. 6D.
- S3: Quality of the westerns needs to be improved. As it stands, there is a wide variation in the expression level of the mutants. Authors need to provide a relative quantification of the expression (with respect to GAPDH). Also, does the marker band on Fig. S3 belong to a different blot? If yes authors should indicate that specifically.
- Is excision activity or DNA integration activity normalized to the expression of proteins?
Reviewer 2 Report
Sharma et al. predicted the structures of human THAP9 protein and the transposase of one family of zebrafish P element (Pdre2), using homology modeling on the solved structure of the transposase protein encoded by the P element from Drosophila melanogaster. Based on the prediction, the authors made a series of mutant proteins to investigate the functional importance of conserved residues. Overall, the study appears basic and did not generate unexpected findings.
My main concern is the reliability of the prediction of catalytic triads in hTHAP9 and the correspondence between residues in hTHAP9 and in DmTNP. Prediction of catalytic triads in hTHAP9 is questionable with the data shown in the current manuscript. The authors claim that D304 in hTHAP9 corresponds to D230 in DmTNP based on their homology modeling, and this assumption is the basis for biochemical assays. However, homology modeling does not indicate the positioning of D304 in the close vicinity of catalytic site. Besides, the multiple alignment (figure 3 and table 1) does not align D304 in hTHAP9 with D230 in DmTNP. It is notable that in the multiple alignment, D230 in DmTNP does not have a corresponding residue in hTHAP9, indicating the loss of catalytic residue in hTHAP9. Given that mutagenesis on several conserved D residues almost eliminate the function of excision or integration, the loss of function in hTHAP9 D304C is not enough to support the identification of D304 as the first D residue of the catalytic triad. If the authors consider the prediction with homology modeling is accurate, they should clarify why it is more reliable than the indication based on the multiple alignment.
Another concern is in the structure of manuscript and some informal description. A certain amount of description in Results should be placed in either Introduction or Materials and Methods. For example, the 2 paragraphs at the beginning of Results should be placed in Introduction. I am noted by some improper text, such as the text in parentheses in p. 9 “(second strand … is octameric)”. The authors should avoid such informal texts in the manuscript.
Minor comments:
p.1
“part of the same”
I am not sure what the authors intend to mean.
p.2 l. 4-6.
These two sentences are inconsistent. Since some eukaryotic TEs have DDD triad in their transposases, the first sentence should be revised.
Figure 1.
Is the lack of alpha3 in Pdre2 predicted? If so, please explain how this difference from the structure of DmTNP contributes to the overall structure and function.
p.6
SETMAR introduced in Introduction and Metnase introduced in the mutagenesis subsection of Results are the identical protein.
p.7. l. 7
“DmTNP superfamily” should be “P superfamily”.
p. 8 bottom
“few LTR transposons”
Basically, all autonomous LTR retrotransposons encode integrases of RNase H fold.
I feel odd that the authors repeatedly use “recent”, “recently solved” for the cryo-EM structure of DmTNP, as the authors use the data in the study from the beginning.
Reviewer 3 Report
Briefly, I think the manuscript is worth publishing in Cells.
Using site-directed mutagenesis, the authors show that the D304 D374 E613 triad of the human domesticated transposase hTHAP9 is essential for excision and transposition activity. However, other conserved D and E residues are also essential, which makes it more complicated to clearly identify the catalytic triad. Thanks to secondary structure predictions they could remove the ambiguity and confirm the identity of the catalytic residues within the RNase H domain of hTHAP9.
Some corrections are necessary:
p. 6 last sentence of section entitled "Site directed mutagenesis..." : E519C should be corrected to D519C in the list of mutations introduced in hTHAP9
The excision assay shown in Figure S5 is confusing. According to the materials and methods, there are more than 2 PvuII sites in the reporter plasmid. They should be indicated in the plasmid maps on the left. On the gels, the bands originating from the reporter plasmid should be labeled (by a dot or asterisk). The reason why there are so many bands in the PvuII digest is not clear: do some of them come from the transposase plasmid?
Round 2
Reviewer 1 Report
The authors addressed all the concerns and the manuscript is improved in quality. I recommend its publication.
Author Response
Thanks very much for your review.Reviewer 2 Report
I consider that the current manuscript is not yet scientifically sound.
The authors PREDICTED the secondary structures of two proteins in the subheading “Prediction of secondary structure elements and RNase-H domain in hTHAP9 and its homologs.” The authors replied that the alpha3 of Pdre2 is predicted inside of ID. It seems true, but the main point is how the lack of alpha helix corresponding to the alpha3 of DmTNP and hTHAP9 affects the overall structure of Pdre2. How do the authors suggest the similarity in structure between DmTNP and Pdre2?
Even though the authors replied “We have not claimed that D304 (of hTHAP9) corresponds to D230 (of DmTNP).”, Fig. 4 indicates the correspondence between D230 of DmTNP and D304 of hTHAP9. In the results of biochemical assay, the authors state “We identify the catalytic domain of hTHAP9 and demonstrate that it has a putative RNase-H like fold (like in DmTNP) with a conserved DDE motif made up of D304, D374 and E613 that constitute the catalytic triad (D230, D303 and E531 in DmTNP)”. On the other hand, Fig. 3 and Table 1 seems to claim the correspondence between D252, instead of D230 of DmTNP and D304 of hTHAP9.
Do the authors claim that D304 of hTHAP9 does not correspond to D230 of DmTNP, but is recruited to constitute one of three catalytic residues after the loss of original catalytic D residue? If the multiple alignment suggested the loss of catalytic D residue in the lineage leading to hTHAP9, K282 of hTHAP9 should correspond to D230 of DmTNP. Even if so, the results shown in the manuscript cannot support their claim.
D252C of DmTNP shows a significant reduction of excision and integration activities. It indicates that even though it is not one of the three catalytic residues, it is essential for the transposase activity. Therefore, the very low excision activity seen with D304C, D374C or D519C of hTHAP9 does not fully justify the authors’ claim that these three residues are the catalytic triad.
Author Response
Reviewer 2
The authors PREDICTED the secondary structures of two proteins in the subheading “Prediction of secondary structure elements and RNase-H domain in hTHAP9 and its homologs.” The authors replied that the alpha3 of Pdre2 is predicted inside of ID. It seems true, but the main point is how the lack of alpha helix corresponding to the alpha3 of DmTNP and hTHAP9 affects the overall structure of Pdre2. How do the authors suggest the similarity in structure between DmTNP and Pdre2?
We did not perform homology modelling of Pdre2 and thus cannot comment on the structural similarity, if any, between DmTNP and Pdre2. We have only performed secondary structure predictions on Pdre2.
Even though the authors replied “We have not claimed that D304 (of hTHAP9) corresponds to D230 (of DmTNP).”, Fig. 4 indicates the correspondence between D230 of DmTNP and D304 of hTHAP9. In the results of biochemical assay, the authors state “We identify the catalytic domain of hTHAP9 and demonstrate that it has a putative RNase-H like fold (like in DmTNP) with a conserved DDE motif made up of D304, D374 and E613 that constitute the catalytic triad (D230, D303 and E531 in DmTNP)”. On the other hand, Fig. 3 and Table 1 seems to claim the correspondence between D252, instead of D230 of DmTNP and D304 of hTHAP9.
Do the authors claim that D304 of hTHAP9 does not correspond to D230 of DmTNP, but is recruited to constitute one of three catalytic residues after the loss of original catalytic D residue? If the multiple alignment suggested the loss of catalytic D residue in the lineage leading to hTHAP9, K282 of hTHAP9 should correspond to D230 of DmTNP. Even if so, the results shown in the manuscript cannot support their claim.
D252C of DmTNP shows a significant reduction of excision and integration activities. It indicates that even though it is not one of the three catalytic residues, it is essential for the transposase activity. Therefore, the very low excision activity seen with D304C, D374C or D519C of hTHAP9 does not fully justify the authors’ claim that these three residues are the catalytic triad.
We have now edited the mentioned text to highlight the possibility that other conserved acidic residues in the catalytic domain maybe part of the triad. It now reads “Thus, we identify the catalytic domain of hTHAP9 and demonstrate that it has a putative RNase-H like fold (like in DmTNP) with several conserved acidic residues including D304, D374 and E613 that may constitute the catalytic triad (D230, D303 and E531 in DmTNP) and are important for DNA excision and integration”. We also note that future structural analysis of hTHAP9 (bound to donor with/without target DNA) would confirm our predictions.
Table 1 lists the residues in the two proteins that align to each other in the MSA (Fig.3). As noted by the reviewer, D252 (DmTNP) aligns with D304 (hTHAP9). However, it is to be noted that the two proteins are of different length (DmTNP 751 residues, hTHAP9 902 residues) and alignment need not suggest that the two amino acids have corresponding functions. Also, we have not claimed that D304 (of hTHAP9) corresponds to D230 of DmTNP. In fact, in a multiple sequence alignment, hTHAP9 has no D/E residue corresponding to D230 position of DmTNP. We made the prediction that D304 may be the first residue of the catalytic triad on the basis of combined results of sequence alignment and structure prediction.
In response to the reviewer’s comment above, as mentioned before, there is evidence that the third residue of a catalytic triad may orient itself in the catalytic pocket after the assembly of the whole transpososome (eg: Rag recombinase). Like we note in the introduction,”Interestingly, this distal position of hTHAP9’s third acidic residue (D304) away from the catalytic pocket resembles the position of E962 in the catalytic triad of Rag1 recombinase (Ref. 27). A conformational change in Rag1 during catalysis brings E962 (third acidic residue) proximal to the catalytic site (Ref. 27).